

# Experimental evidences for bifurcation angles control on abandoned channel fill geometry

Léo Szewczyk[1], Jean-Louis Grimaud[1], Isabelle Cojan[1]

[1]MINES ParisTech, PSl Research University, Centre de Géosciences, 35 rue St Honoré 77305 Fontainebleau Cedex, France

*Correspondence to*: Léo Szewczyk (leo.szewczyk@mines-paristech.fr)

**Abstract.** The nature of abandoned channels sedimentary fills has a significant influence on the development and evolution of floodplains and ultimately on fluvial reservoir geometry. A control of bifurcation geometry (i.e., bifurcation angle) on channel abandonment dynamics and resulting channel fills, such as sandplug, has been intuited many times but never quantified. In this study we present a series of experiments focusing on bedload transport designed to test the conditions for
channel abandonment by modifying the bifurcation angle between channels, the flow incidence angles and the differential channel bottom slopes. We find that disconnection is possible in the case of asymmetrical bifurcations with high diversion angle (≥30°) and quantify for the first time a relationship between diversion angle and sandplug length and volume. The resulting sandplug formation is initiated in the flow separation zone at the external bank of the mouth of the diverted channel. Sedimentation in this zone initiates a feedback loop leading to sandplug growth, discharge decrease and eventually
to channel disconnection. Finally, the formation processes and final complex architecture of sandplugs are described, allowing for a better understanding of their geometry. Although our setup lacks the complexity of natural rivers, our results seem to apply at larger scales. Taken into account, these new data will improve the realism of fluvial models.

## 1 Introduction

Abandoned channels are ubiquitous features of the alluvial plain, which have a huge impact on the fluvial system evolution
and properties. First, abandoned channels form local topographic lows that trap sediments (Aalto et al., 2008; Lauer & Parker, 2008; Dieras et al., 2013) and host wetlands (Novitsky et al., 1996; Ward et al., 1999). Second, the fine grain fraction of their filling may influence active channels migration, as clays are more resistant to erosion than sandy sediments (Howard 1992; Smith et al., 1998; Berendsen & Stouthamer, 2000; Schwendel et al., 2015). Last, abandoned channels are filled with sediments of varied permeability, which may impact flow path in active alluvial plains (Flipo et al., 2014) and ultimately in
the resulting geological reservoir (Miall, 1996; Willis & Tang, 2010; Colombera et al., 2017; Cabello et al., 2018). Indeed, recent studies have shown that sedimentary fills are complex bodies and may contain coarser sediments than initially assumed (Hooke, 1995; Toonen et al, 2012; Dieras et al., 2013). When integrated to reservoir flow simulations, these coarse deposits may drastically change the connectivity of otherwise isolated sand bodies (e.g., point-bars; Donselaar & Overdeem, 2008).



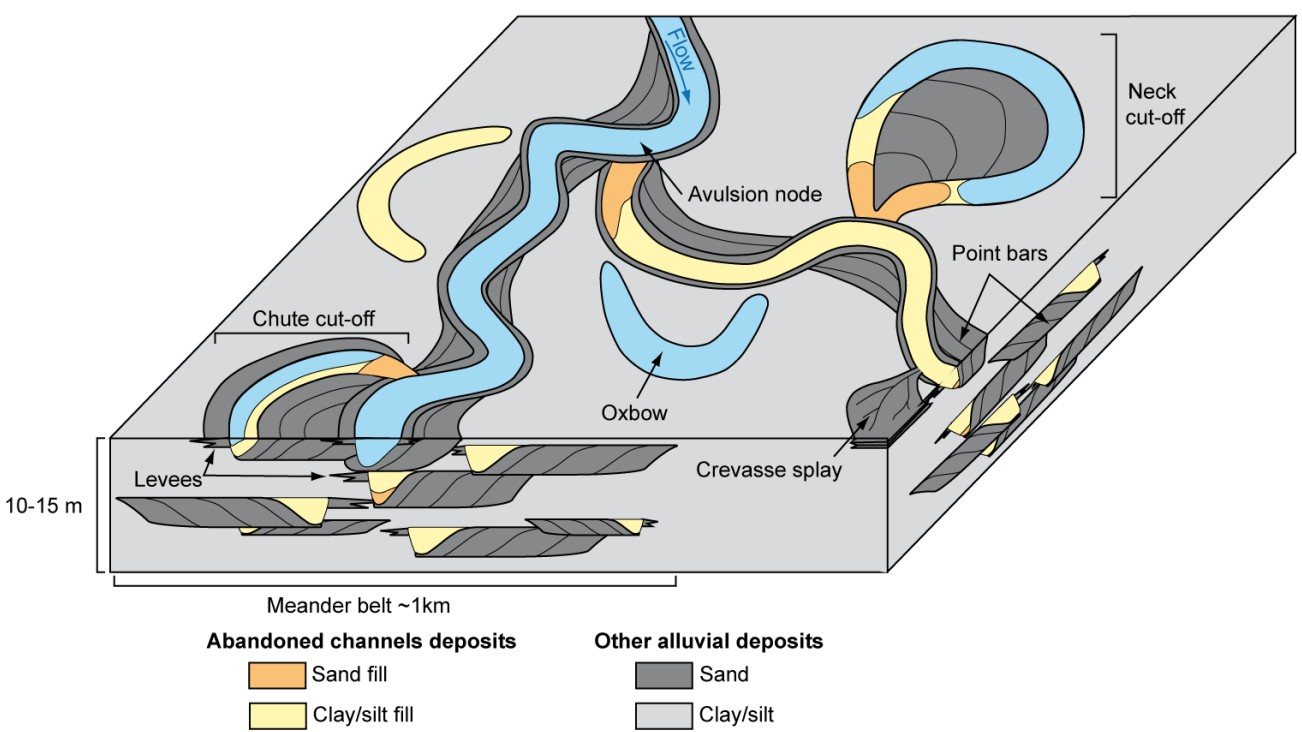

**Figure 1: 3D sketch showing the occurrences of deposits associated with abandoned channels in an alluvial plain.**

Currently, abandoned channels are studied on the field (Hooke, 1995; Constantine et al., 2010; Dieras et al., 2013) but less so in numerical models and experiments. Different styles of abandonment are observed in fluvial systems (i.e., cut-offs, avulsions), implying the formation of sedimentary fills of various grain-sizes and geometries (Allen, 1965; Toonen et al.,

2012; Fig. 1). A common thread to existing models is that abandonment is the consequence of the formation of a wedge-shaped sandplug in one of two channels shortly after a bifurcation (Fig. 1). Disconnected channels are then mostly filled by fine-grained overbank flood sediment (Bridge et al., 1986; Plint, 1995; Bridge, 2003). The coarse deposits are introduced beforehand as bedload, i.e., as long as there is a connection with the active channel. The dynamics at the bifurcation during the disconnection phase have therefore a key control on the sediment architecture of later abandoned channels (Bertoldi,

2012; Bolla Pittaluga et al., 2015; Constanttine et al., 2010; Kleinhans et al., 2013).

Based on field studies, the geometry of the bifurcation, particularly the upstream bifurcation angle, is thought to control the duration of (dis)connection and therefore sandplug accretion and geometry (Fisk, 1947; Shields et al., 1984; Shields & Abt, 1989), but most authors agree that bifurcations remain overlooked in alluvial plains (Constantine et al., 2010; Kleinhans et al., 2013).

Existing numerical and experimental studies focus on the parameters controlling discharge and sediment partitioning at bifurcation (Bulle, 1926; de Heer & Mosselman, 2004; Kleinhans et al., 2008, 2013; Salter et al., 2018, 2019) and bifurcation (un)stability (Bertoldi & Tubino, 2007; Bolla Pittaluga et al., 2003, 2015; Iwantoro et al., in review). To our



knowledge, no study currently exists that focuses specifically on quantifying the condition(s) for abandoning channels at a bifurcation and on the resulting sediment architecture.

In this work, we study experimentally and quantify for the first time the influence of bifurcation geometry, specifically the diversion angle, on fluvial channel abandonment. We focus on (1) abandonment potential and the associated processes and (2) the extent and geometry of the sedimentary bodies formed by bedload deposition in abandoned channels, i.e., sand-plugs and sand-bars.

## 2 Methods

### 2.1 Experimental design

Experiments were carried out in the Geomorphic Lab of the Centre de Géosciences of MINES Paristech, Fontainebleau. A modular flume with fixed walls was built. It was composed of three branches: one inlet and two distributary channels connected through a bifurcation area (Fig. 2). Each channel had a width $W$ of 4 cm and a length of about 75 cm. The global slope of the experiment was 1.48 % while the slopes in distributaries 1 and 2, respectively $S_1$ and $S_2$, varied with the
configuration (Table 1). The bifurcation area was modular, to allow different angles between the inlet and the distributary channels. Three angles were considered (Table 1, Fig. 2). The bifurcation angle $\alpha$ was the angle between the two distributaries. The incidence angles $\beta_1$ and $\beta_2$ were the angles between the inlet channel and the stream-left and -right distributaries, respectively (Fig. 2). When $\beta_1 = 0$, then $\beta_2 = \alpha$ and corresponded to a diversion angle, as usually defined.

The experiments started in an empty flume and typically lasted 90 to 100 minutes. Input water and sediment discharges were
constantly fed at rates of 300 L h$^{-1}$ and 0.6 L h$^{-1}$, respectively. Both fluxes were calibrated beforehand to allow formation of sedimentary structures without filling the flume too quickly, allowing observations to be made. The water was delivered through a head tank to reduce turbulence in the incoming flow. The water was dyed in blue using food colorant in order to enhance contrast in pictures.





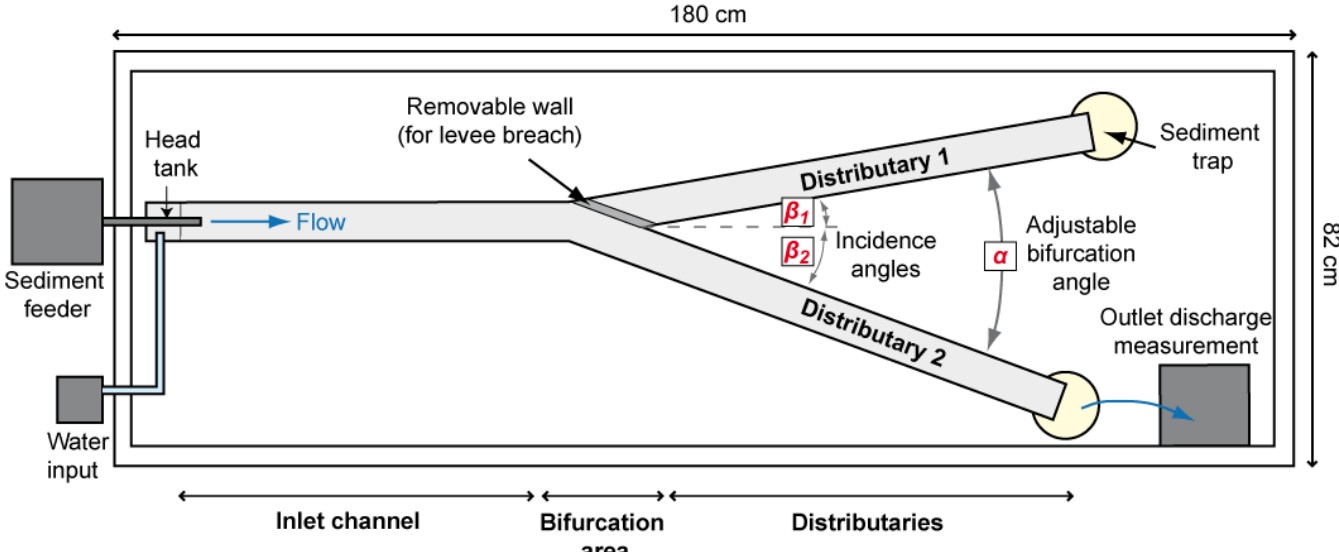

**Figure 2: Overhead view of the experimental setup together with the different angles considered and the levee breach setup.**

Sixteen experiments were designed to explore the influence of bifurcation angle $\alpha$ values ranging from 30 to 90° and $\beta_1$ and associated $\beta_2$ values ranging from 0 to 90° (Exp. 1 to 16) (Table 1). A first set of 4 symmetrical ($\beta_1=\beta_2=1/2\alpha$, Exp. 3, 7, 11 and 14) (Fig. 3a) and 5 asymmetrical ($\beta_1 \neq \beta_2$, Exp. 1, 5, 9, 12 and 15) (Fig. 3b) configurations were built. A second set of 7 experiments (Exp. 2, 4, 6, 8, 10, 13 and 16) replicated the configurations of the first set (except for Exp. 11 and 14) with the

addition of a removable wall placed at the entrance of distributary 1 parallel to the orientation of Distributary 2(Figs. 2 and 3c) (Supplementary Table). The wall was removed after the system had reached equilibrium (identical input and output sediment discharges) to simulate a levee breach (Fig. 3c). In all asymmetrical configurations (except for Exp. 1 and 2), distributary 1 was straight ($\beta_1=0$) so that $\beta_2$ was a diversion angle (Table 1).

A final set of 3 experiments (Exp. 17, 18 and 19) was designed to determine if the observed effects of a given diversion

angle could be counterbalanced by a slope variation in the deviated distributary. The experiments had the same planar geometry than Exp. 15 ($\alpha = \beta_2 = 90°$, slope ratio $S_2/S_1 = 0$), but the slope at the bottom of distributary 2 was modified so that 3 additional different slope ratios $S_2/S_1$ could be tested (i.e., respectively 0.68, 0.87 and 1.73 in Exp. 17, 18 and 19).



| Experiment | Bifurcation angle α (°) | Distributary 1 | | | | Distributary 2 | | | |
|---|---|---|---|---|---|---|---|---|---|
| | | β1 (°) | Bed Slope S1 (%) | Equilibrium water discharge (L.h⁻¹) | Shields parameter θ | β2 (°) | Bed Slope S2 (%) | Equilibrium water discharge (L.h⁻¹) | Shields parameter θ |
| 1 | 30 | 10 | 1.468 | - | 0.213 | 20 | 1.418 | - | 0.123 |
| 2 | | | | - | 0.213 | | | - | 0.082 |
| 3 | | 15 | 1.443 | 153.4 | 0.209 | 15 | 1.443 | 146.6 | 0.209 |
| 4 | | | | 150.6 | 0.209 | | | 149.4 | 0.084 |
| 5 | | 0 | 1.48 | 250 | 0.172 | 30 | 1.312 | 50 | 0.114 |
| 6 | | | | 262 | 0.214 | | | 38 | 0.076 |
| 7 | 45 | 22.5 | 1.405 | 150.6 | 0.204 | 22.5 | 1.405 | 149.4 | 0.204 |
| 8 | | | | 161.4 | 0.204 | | | 138.6 | 0.244 |
| 9 | | 0 | 1.48 | 299 | 0.172 | 45 | 1.193 | 1 | 0.138 |
| 10 | | | | 274.2 | 0.214 | | | 25.8 | 0.069 |
| 11 | 60 | 30 | 1.312 | 181.5 | 0.190 | 30 | 1.312 | 118.5 | 0.114 |
| 12 | | 0 | 1.48 | 270.1 | 0.214 | 60 | 0.98 | 29.9 | 0.085 |
| 13 | | | | 294.3 | 0.214 | | | 5.7 | 0.085 |
| 14 | 90 | 45 | 1.193 | 193.6 | 0.138 | 45 | 1.193 | 106.4 | 0.207 |
| 15 | | 0 | 1.48 | 294.3 | 0.172 | 90 | 0 | 5.7 | 0.116 |
| 16 | | | | 258 | 0.214 | | | 42 | 0.058 |
| 17 | | | | 299 | 0.214 | | 1 | 1 | 0.087 |
| 18 | | | | 294.3 | 0.214 | | 1.282 | 5.7 | 0.149 |
| 19 | | | | 291.6 | 0.214 | | 2.56 | 8.4 | 0.074 |

**Table 1: List of the conducted experiments and associated parameters.**

## 2.2 Data acquisition

Free surface elevation was periodically measured in all channels. Water discharge was measured out of the two distributaries using a system similar to that of Salter et al. (2019). Water was flowing out of the channel into a cylinder with a hole at the bottom small enough to allow variation of the level in the cylinder. The weight evolution was measured within the cylinder using a digital scale and converted into discharge using a calibration curve. Figures 3d, 3e, 3f and 3g show the resulting water discharge partitioning for experiments without removable wall. The sediment was a well-sorted, rounded to sub-angular, fine ($d_{50}$=209 µm) Fontainebleau sand. Sediment traps allowed quantifying the volume of sediment that bypassed the distributaries.

Pictures of the flume were taken every minute by an overhead camera to observe sandbodies' formation and measure their length. Sandbody's total length, i.e., including both subaerial and submarine parts, was measured from pictures. Sandplug construction was reported by increments of 15 minutes for asymmetrical configurations without levee breach (Fig. 4). Sandplug length was measured at the last location where it extended over the whole channel width and at its downstream limit. In the following, the mean of these two measurements is used to speak of sandplug length (Fig. 4). The final Digital Elevation Models (DEMs) of the deposits were computed from 3D photogrammetric surveys taken by two cameras mounted on a mobile rail using the Agisoft PhotoScan Professional v1.4 software (Fig. 5). The DEMs (precision of 0.4 to 0.5 mm) were used to produce mean longitudinal elevation profiles and to measure the longitudinal slope of sediment deposited in disconnected channel 2 (Fig. 6). Finally, sandplug volume calculated from DEMs and sandplug length $L$ -divided by channel width $W$- were compared to bifurcation angles and slope ratios (Fig. 7).





### 2.3 Sediment transport law

A sediment transport law was calibrated to compare the experimental results with theory using the methods of Seizilles

(2013) and Delorme et al., (2017). A series of runs were carried out with constant water and sediment feed rates in a 3cm-wide flume. The experiment was repeated with different sediment discharges $Q_s$ and the equilibrium slope was measured each time after 10-15 hour to estimate the associated Shields parameter $\theta$ (Fig. 8a). A critical Shields parameter $\theta_c$ -where a threshold of motion (i.e., $q_s = Q_s/W > 0$ g s$^{-1}$ m$^{-1}$) was attained (Fig. 8a)- was estimated using:

$$\frac{Q_s}{W} = q_0 \left( \theta - \theta_c \right)$$  Eq. 1

The critical Shields parameter was then used to estimate channel slope at threshold (see Métivier et al., 2017 and references therein):

$$S = \left[ \sqrt{\mu} \left( \frac{\theta_c (\rho_s - \rho)}{\rho} \right)^{\frac{5}{4}} \sqrt{\frac{\kappa\left[\frac{1}{2}\right] 2^{\frac{2}{3}}}{3 C_f}} \right] Q_*^{-\frac{1}{2}}$$  Eq. 2

where $Q_* = Q/\sqrt{g d_s^5}$ is the dimensionless discharge, $\rho$ and $\rho_s$ the densities of water and sediment, $\mu$ the friction angle, $C_f$ the turbulent friction coefficient and $\kappa[1/2]\approx1.85$ a transcendental integral. In our setup, $\rho = 1000$ kg m$^{-3}$, $\rho_s = 2650$ kg m$^{-3}$ and $\mu$

$\approx 0.7$. $C_f$ varied between 0.02 and 0.1 as an uncertainty estimate (Fig. 8b).

In experiments where a disconnection was observed, $Q$ values were measured 2 minutes before disconnection to compare sandplug slope with $Q_*$ according to the threshold theory (Fig. 8b). Finally, a run was performed for about ten hours in a 4 cm-wide flume with a 1.48% bottom slope to mimic the inlet channel conditions. The resulting bed slope was compared to the theory (Fig. 8b).

**3 Results**

Each experiment began with a short (15-20 minutes) phase of progradation of the sediment down to the bifurcation point. The sediment started forming a sandbar downstream of the sediment feeder and then split into alternate bars that migrated through the inlet channel. Once sediment reached the bifurcation, it was partitioned into the distributaries, except when a removable wall was present. After an adjustment period, water discharge was considered at equilibrium when it remained

constant -or slightly varied around a constant value- in each channel (Fig. 3). The associated water partitioning could be *equal* (i.e., identical in both branches) or *unequal* (i.e., different discharges in both branches). Sediment bypassing each distributary channel at equilibrium was roughly proportional to discharge partitioning. A distributary channel was considered disconnected when no bedload movement was observed. Usually, such disconnection occurred before water discharge equilibrium was attained (Figs. 3f and 3g).



### 3.1 Bifurcation geometries allowing channel disconnection

In the case of symmetrical bifurcations (Figs. 3a, 3d and 3e), no disconnection occurred. For low bifurcation angle ($\alpha \leq 45°$, Exp. 3 and 7), a stage of soft avulsions (*sensus* Salter et al. (2018)) was observed initially (Fig. 3d). In this case, equilibrium with equal water partitioning was attained after both distributaries had been filled with sediment.

For high bifurcation angles ($\alpha \geq 60°$, Exp. 11 and 14), an unequal water discharge partitioning lasted since the beginning of the experiment (Fig. 3e) and no soft avulsion stage was observed. The degree of such discharge asymmetry increased with bifurcation angle (Fig. 3e). Discharge partitioning in replicates of experimental runs 11 and 14 showed that the favored distributary channel was randomly chosen due to the interaction between the flow, the upstream alternate bars and the wedge of the bifurcation point. This indicated that the unequal water partitioning was not due to a tilt in the experimental setup.

All asymmetrical configurations reached equilibrium with unequal water discharge partitioning (Figs. 3f and 3g). In Exp 1 and 2 ($\beta_1=10°$, $\beta_2=20°$), no disconnection was observed. In the other asymmetrical experiments (i.e., Exp 3, 4, 9, 10, 12, 13 and 15 to 19), disconnection was always attained and a sandplug formed (Figs. 3b and 3c). Water discharge partitioning was unequal between the two distributaries. Then, as a sandplug started to form in distributary channel 2 while the straight distributary ($\beta_1 = 0$) captured an increasingly important portion of the flow (Fig. 3f). Discharge asymmetry observed at equilibrium was globally proportional to $\beta_2$. In Exp. 9 ($\beta_2 = 45°$) disconnection was reached very quickly (50 minutes) due to the development of an additional sandbar on top of the sandplug that steered the flow towards the first distributary (Fig.3f). This was the only occurrence of such a phenomenon out of the 19 experiments. In all experiments, fluctuations of water discharge were observed. Visual inspections showed that these fluctuations were linked to the migration of alternate bars in the inlet channel (Bertoldi & Tubino, 2007). A bar located stream-right immediately before the bifurcation provided sediment to the distributary 2 and temporarily decreased its water discharge, favoring deposition and construction of the sand plug. A bar located stream-left immediately before the bifurcation temporarily decreased sediment supply and increased water discharge to distributary 2, favoring sediment entrainment.

Levee breach experiments reached final equilibrium 5 to 10 minutes after distributary 1 was opened. Their final equilibrium state was very similar to that of the experiments made without levee breach in the same configuration (Figs. 3b and 3c). In asymmetrical experiments with levee breach (Exp. 6, 10, 13 and 16), sediment transited through distributary 2 before the opening of distributary 1. This allowed the deposition of alternate bars (Fig. 3c) in distributary 2. In the asymmetrical experiments without levee breach (i.e., Exp 5, 9, 12, 15, 17, 18 and 19), all sediment that entered distributary 2 was deposited to form a sandplug (Figs. 3b and 5b).

Figure 3g shows discharge partitioning under varying slope ratios $S_2 / S_1$ (0, 0.68, 0.87 and 1.73) for the same bifurcation angles ($\beta_1=0$, $\alpha=\beta_2=90°$). In each case, the same final steady state ($Q_{w1}\approx290$ L h$^{-1}$, $Q_{w2}\approx10$ L h$^{-1}$) was reached and distributary channel 2 disconnected (Fig. 3g). A sandplug thus formed, albeit at different speeds. Overall, the time required to reach disconnection increased with slope ratio (Fig. 3g). Exp. 15 ($S_2/S_1 = 0$) did not strictly follow the trend and showed the slowest sandplug building rate of the four experiments.



**Figure 3: Evolution of the experiments. (a-c) Overhead pictures of the setup for symmetrical (a) and asymmetrical configurations**
**without (b) and with levee breach (b). (d-f) Evolution of water discharge measurement at the output of the distributary channels**
**for symmetrical (d-e) and asymmetrical (f-g) configurations. (g) Asymmetrical 90° configurations with varying slope ratios $S_2 / S_1$.**





### 3.2 Sandplug formation dynamics and architecture

During sandplug formation, discharge gradually decreased in distributary channel 2, until a constant value was attained (Fig. 3f) and no more sediment motion was observed. Sandplug growth processes and final architecture were very similar in all
these experiments, regardless of the incidence angle (Fig. 4).

The formation of the sandplug initiated on the external side of the diverted channel, immediately after the bifurcation (Fig. 4). In this area flow velocity was the lowest, allowing the deposition of sand that initiated the first fixed sandbar. This first bar anchored to the external bank of the channel and quickly grew downstream and towards channel centerline (Fig. 4a). It then widened and lengthened until its growth stopped. Other bars then formed from the sides of previously formed
sandbar(s) and stretched downstream, resulting in a composite sandplug (Figs. 4a and 8). When sand was deposited over the entire width of the channel, a thalweg formed across the sandplug. It allowed sediment transfer downstream until it was buried by a sandbar that disconnected the whole channel. Overall, a slight decrease of sandplug growth speed with diversion angle is observed.

In the case of levee breach experiments (Exp. 6, 10, 13 and 16), alternate bars formed in distributary 2 before the opening of
distributary 1 (Fig. 3c). After levee breach, the discharge abruptly decreased in distributary 2, initiating the rapid (2-5 minutes) formation of the sandplug. In the meantime a transient knickpoint formed along the inlet channel to allow for bed slope adjustment. Parts of the existing alternate bars at the bifurcation were thus reworked (Fig. 8). A small plug, mostly consisting of the reworked deposits, formed rapidly at the entrance of distributary channel 2.

Sandplug long slope increased with the diversion angle (from 2.5% to 7.8%; Table 1). This was the most visible in the no
forcing scenarii, as there was no interference from previously deposited alternate bars (Fig. 6a). Bars after the plug itself showed the shallower slopes (Fig. 6b). In the case of levee breach, the sandplug slope was slightly steeper when the diversion angle was lower or equal to 45 (Supplementary Table 1). Figure 8b shows that sandplug slopes are overall consistent with the threshold theory when compared to the dimensionless discharge, with deviations towards lower or steeper slopes than the theory depending on the forcing scenario. The upstream end of the sandplug was about 4 to 8 mm more
elevated than the active channel bed (Figs. 5a and 6a) and was slightly elevated compared to the water elevation in the active channel, similarly to a levee in natural systems. In the case of levee breach the inherited geometry caused by the alternate bars is preserved, resulting in a sandplug with a roughly equal maximum elevation but a higher overall elevation (Fig. 6b).





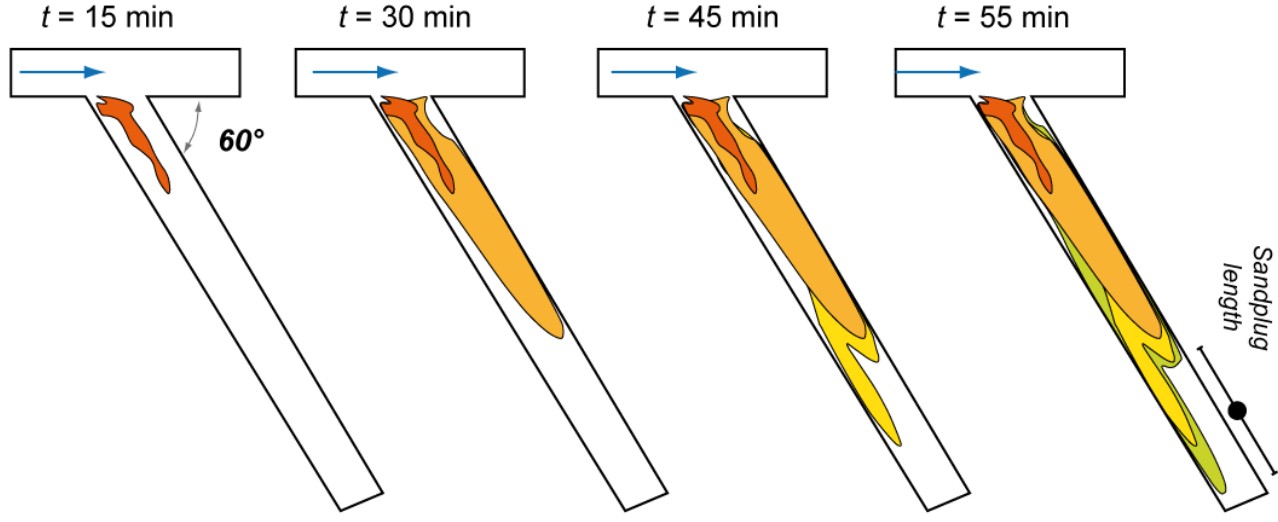

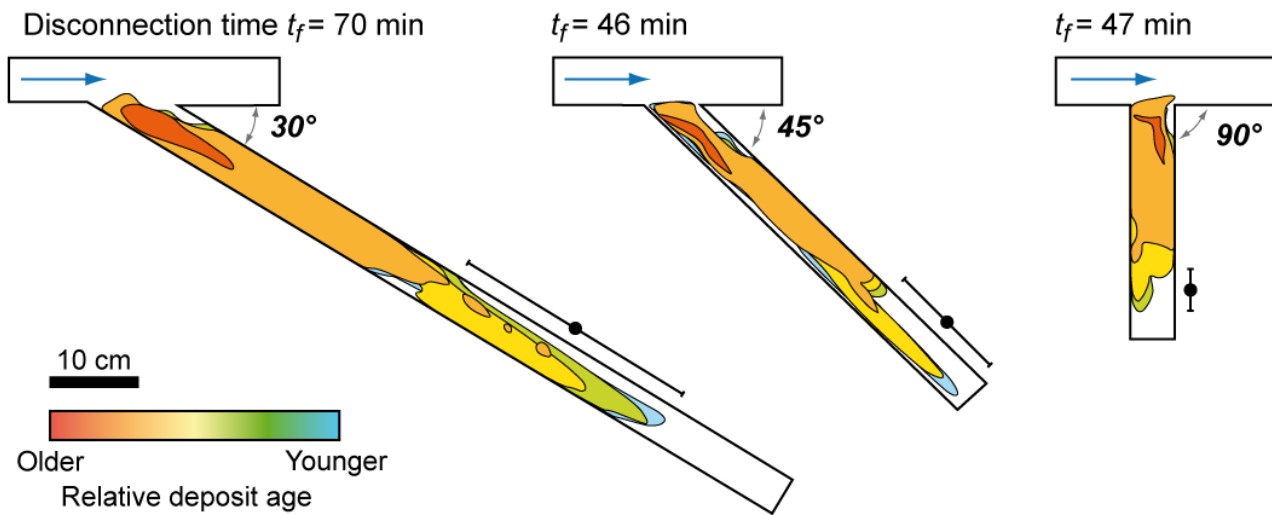

**Figure 4: Planar growth of sandplug from overhead pictures. (a) Consecutive growth phases of the sandplug for $\beta_2$=30°. (b) Final states of the sandplug for different $\beta_2$ angles. Contours are drawn every 15 minutes. The mean sandplug length and uncertainties is represented with a black circle.**




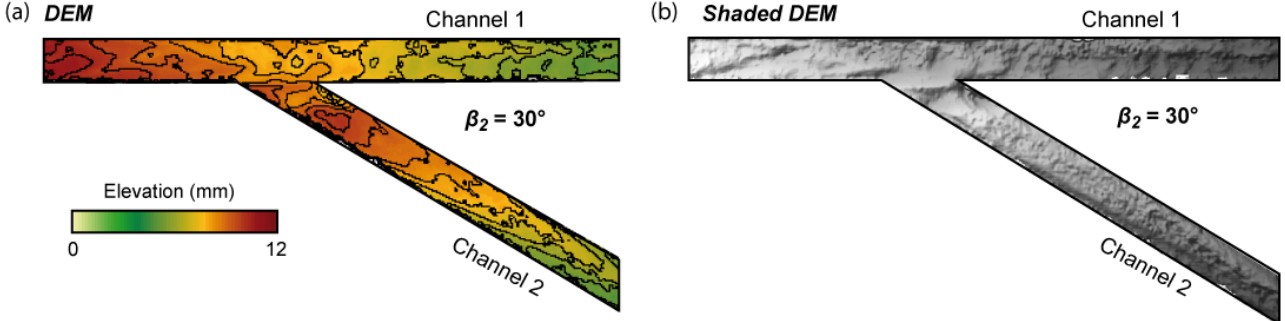

**Figure 5: Final topography of the Experiment 5 showing the active (1) and disconnected (2) channels. (a) Elevation DEM. (b) Shaded relief of the same DEM. Contour lines represents 1.5 mm elevation.**

## 3.3 Controls on sandplug length and volume

Relationships between incidence angle $\beta_2$ and sandplug length and volume, as well as between differential bottom slope and sandplug length and volume were found in asymmetrical configurations (Fig. 7). Above a 30° threshold value of $\beta_2$ required to form a sandplug, sandplug length linearly decreased with $\beta_2$ (Fig. 7a). In case of levee breach, sandplugs were shorter for a given $\beta_2$ value (Fig. 7a). In fact, the regression lines calculated in the cases without and with levee breach returned coefficient values of about -0.11 and -0.08, respectively. This induces a slow convergence of sandplug lengths towards higher angles and shows that the sandplugs formed by a levee breach are slightly less affected by the $\beta_2$ value.

Without levee breach, the sandplug was the only bedload deposit in the disconnected distributary (Exp. 5, 9, 12 and 15) (Figs. 3b and 5b). Its volume steadily decreased with $\beta_2$, following a roughly linear relationship (Fig. 7b). With a levee breach (Exp. 6, 10, 13 and 16), the total volume of sediment also decreased linearly with $\beta_2$, but was globally higher. Indeed, in the levee breach experiments the sandplug accounts for only part of the sand volume deposited in the distributary, the rest being accounted for by the alternate bars formed during the channel active phase. When only the volume of the sandplug itself is taken into account, sandplug volume was thus lower than in the no forcing cases (Fig. 7b). Figure 7c indicates that the length and volume of the sandplug was also slightly affected by the slope ratio. The sandplug became more elongated and thinner as the slope increased in distributary 2 and consequently sandplug slope overall decreased with an increase of the slope ratio, from 0.68-0.87 (Exp. 15 and 17) to 1.73 (Exp. 19).





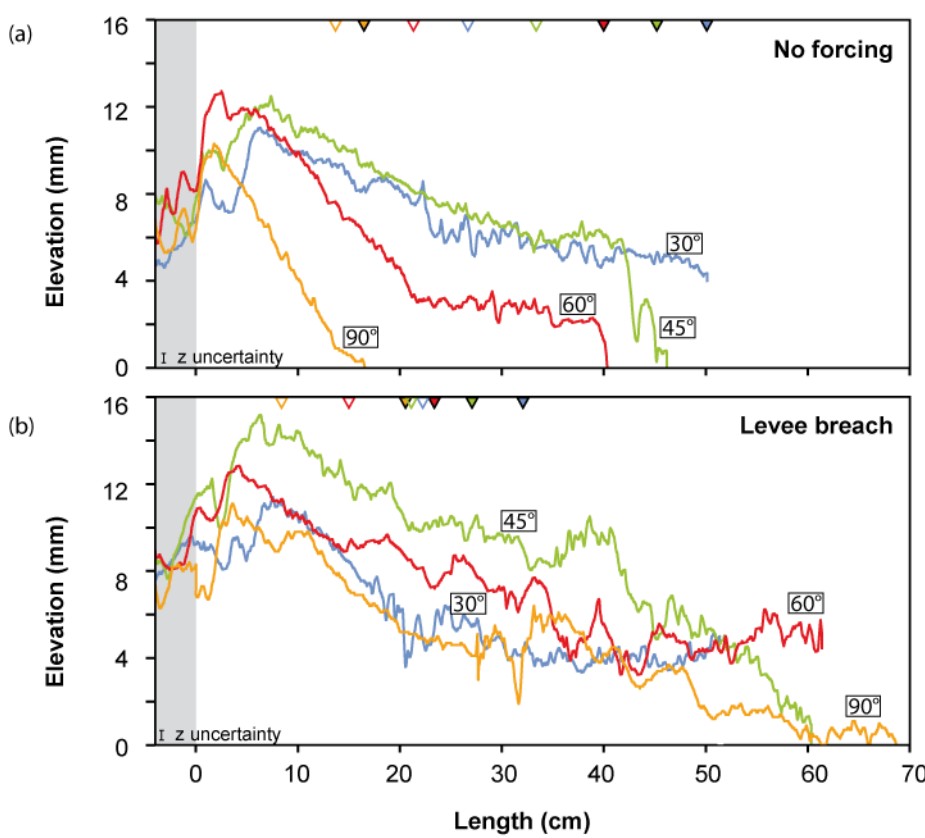

**Figure 6: Mean longitudinal elevation profiles of the sandplugs for increasing $\beta_2$ angles without (a) and with levee breach (b). Profiles were corrected for bottom slope to facilitate comparisons. The grey area corresponds to the active channel. Empty triangles correspond to the furthest location where the sandplug occupies the entire channel width while full triangles indicate the extremity of the sandplug.**








Figure 7: Relationships between sandplug length and volume, and the incidence angle $\beta_2$ (a, b) and slope ratio $S_2/S_1$ for $\beta_2=90°$ (c). Regression line for total sediment in (c) was not calculated since only 3 points were aligned.





## 4 Discussion

### 4.1 Bifurcation angle control on abandonment

In this study, channel disconnection was possible for highly asymmetrical bifurcation (i.e., $\alpha = \beta_2$ and $\beta_2 \geq 30°$) (Exp 5, 6, 9, 10, 12, 13 and 15 to 18). In symmetrical configurations, no disconnection was observed although an effect on discharge partitioning occurred. Such discharge imbalance can be attributed to the bifurcation angle value alone, since the slopes of both distributaries are equal. Exp. 1 and 2 demonstrated that even in asymmetrical cases, no disconnection is possible if the

diversion angle is too low ($\beta_1$, $\beta_2 < 30°$) or the bifurcation not asymmetrical enough. Hence, bifurcation geometry indeed controls the disconnection of channel as proposed by Fisk (1947), Shields et al. (1984) and Shields & Abt (1989). Moreover, our results suggest that disconnection is more sensitive to incidence angle value than bifurcation angle value alone. In essence, the value of the flow deviation from its original trajectory has more impact than the value of flow divergence between the distributaries. The fact that disconnection was observed despite slope advantage in the diverted distributary

channel (Figs. 3g and 7c) (Table 1) is another strong argument in favor of planar geometry control on bifurcation (dis)equilibrium.

In symmetrical experiments, water discharge shifted from a soft avulsion regime with roughly equal partitioning (e.g., Salter et al., 2018) to an unequal partitioning as bifurcation angle $\alpha$ increased (Figs. 3d and 3e). Considering the trend in Figs. 3d and 3e, we may hypothesis that abandonment would have occurred at some point if the bifurcation angle $\alpha$ had been

increased further. In a comparable set of experiments, Bertoldi and Tubino (2007) and Salter et al. (2019) showed that disconnection may also be possible for low-angle symmetrical configurations (respectively 30 and 16°). Their work focused on delta networks, where bifurcations can be considered stable with two active distributaries of a width inferior to that of the upstream channel. In their experiments, Bertoldi et al. (2007) and Salter et al. (2019) respectively had a ratio of 0.66 and 0.7. In that case, downstream migration of an alternate bar at equilibrium with the upstream channel to a narrower distributary

may easily lead to sandplug formation. In our fluvial channel focused experiments, where an avulsion generally leads to channel abandonment in favor of the other, the ratio between the width of each distributary and the width of the upstream channel is 1 (Fig. 2). The distributaries could accommodate the alternate bars migrations, preventing disconnection. Furthermore, these authors demonstrated that discharge asymmetry is largely a function of flow aspect ratio upstream of the bifurcation. In our experiment, upstream water depth was usually 4 mm thus aspect ratio was about 10, which is propitious to

equal flow partitioning in symmetrical bifurcation geometry (Salter et al., 2018).

### 4.2 Bifurcation angle control on sandplug extent

Numerous field studies intuited a relationship between the incidence angle and the length of the sandplug, stating that a low incidence angle produced a longer sandplug (Fisk, 1947; Allen, 1965; Gagliano & Howard, 1984; Shields et al., 1984; Shields & Abt, 1989; Dieras et al., 2013). This was explained by the fact that bedload is easily diverted into the channel at

low angles, and on a longer distance before deposition. With increasing bifurcation angles, a smaller fraction of the bedload





enters the channel, resulting in shorter and smaller sandplugs. Our experiments show that bedload partitioning is proportional to discharge partitioning, and that final discharge partitioning is controlled by the diversion angle (Fig. 3). As a result, sandplug volumes and lengths linearly decrease with diversion angle (Figs. 7a-b). Sandplug volumes and lengths are also modulated by the slope of the abandoned channel (Fig. 7c). Sudden events such as levee breaching create shorter and less

voluminous sandplugs, as less bedload material is mobilized to build the sandplug due to the faster disconnection (Figs. 7a and 7b).

The fast disconnection and limited sandplug length and volume observed in case of levee breach (Figs. 7a and 7b) can be explained by the rapid entrenchment of the flow in channel 1. The newly opened channel 1 has a slope advantage as no aggradation occurred during the first phase of the experiment. As the result of incision and knickpoint retreat, a threshold

that prevents part of the bedload to enter distributary 2 is created (Slingerland & Smith, 2004) and the flow is preferentially funneled in the distributary channel 1. It is worth noting that the effect of levee breaching on sandplug length and volume decreases slightly when the incidence angle increases (Figs. 7a and 7b), with a slight convergence towards higher angles. The slight difference in regression lines values between the no forcing and levee breach scenarii also hints that although the sandplug extent and volume are controlled by the diversion angle, different triggers might initiate sandplug formation

depending on the scenario.

As bedload partitioning varies with discharge partitioning in our setup, a smaller fraction of bedload enters the deviated distributary and this fraction diminishes in time with discharge, until disconnection. In a flume with comparable geometry, Bulle (1926) measured of the bedload repartition in distributaries and found that around 90% of the bedload was steered in the deviated distributary for angles ranging from 30 to 150°. However this was at the cost of maintaining discharge equal in

both distributaries, by adjusting water slope using sluice gate elevation at the end of each distributary (Bulle, 1926; Lindner, 1953). The resulting water slope ratios were likely far above the bottom ones we used since we did not observed equal discharge partitioning (i.e., Fig. 3g). Hence, a comparison of sediment repartition between Bulle (1926) and the present work does not seem relevant.

### 4.3 Mechanism for channel abandonment

Beyond the influence of the bifurcation angle, other mechanisms have been invoked to explain channel disconnection: the Shields number of the system upstream of the bifurcation and its effects on the aspect ratio (Wang et al., 1995; Bertoldi et al., 2009; Bolla Pittaluga et al, 2015) and bars presence (Bertoldi & Tubino, 2007; Bertoldi, 2012); as well as the sinuosity upstream of the bifurcation and the slopes in each distributaries (Kleinhans et al, 2013; Van Dijk et al., 2014). Based on the results of this study, one may argue that it is the difference of channel bedslope associated to the bifurcation geometry – and

not the bifurcation geometry itself – that leads to abandonment. The experiments presented in Fig. 3g show that this is not the case in the experiments with asymmetrical bifurcations as a high diversion angle is a sufficient condition for hydraulic disconnection. Having a *reasonable* slope advantage to the diverted channel does not reverse the final outcome; it only affects the time needed to disconnect the channel and the extent of the deposits.



Another proposed mechanism is that low discharge in one channel induces sedimentation, which would in turn further reduce the discharge in the channel by plugging it or by changing the bed slope, creating a feedback loop leading to disconnection (Zolezzi et al., 2006; Bertoldi, 2012). Based on this study, this is discounted as abandonment would have been observed when discharge repartition was unequal from the beginning in symmetrical configurations with incidence angles above the incidence angle 30° threshold (Fig. 3e). The sandplug does create a feedback loop that helps disconnection, but the timing of the discharge decrease in distributary 2 during the asymmetrical experiments shows that it is not the discharge imbalance that initiates the sandplug formation. It is rather the formation of the sandplug that induces the change in discharge partitioning (Fig. 3f), leading to disconnection.

Eddies were observed on the external bank of the channel, immediately downstream of the bifurcation, in which water was slowed and sediment deposited. Bulle (1926) observed that these eddies were produced by flow separation at the bifurcation. These eddies were also found in numerical modeling by de Heer & Mosselman (2004) and van der Mark & Mosselman (2012), and observed by Constantine et al. (2010) on the Sacramento River, and were named *flow separation zone*. The latter authors observed that the width of the flow separation zone increases with the incidence angle value. In our experiments, the wider flow separation zone in the diverted channel led to the formation of the first stage of the sandplug against the external bank of the diverted channel just downstream of the bifurcation (Fig. 4). This in turn led to a further reduction of the flow diverted in the channel, favoring deposition and growth of the sandplug, until the channel got disconnected. At low values of $\beta$, this flow separation zone was too narrow to initiate the feedback loop. When $\beta$ reached 30° it became wide enough to create a larger and higher sandbar and start the feedback loop.

## 4.4 Comparison with field cases and upscaling

Although our model represents a very simplified setup, it has been purpose-built to study the influence of bifurcation geometry on disconnection and on bedload deposits repartition and geometry in abandoned channels. As such it complements previous studies focusing on equilibrium configurations relative to water and sediment flows (Bolla Pittaluga et al., 2003, 2015; Bertoldi & Tubino, 2007; Edmonds & Slingerland, 2008, Bertoldi et al., 2009; Salter et al., 2018, 2019). River channel section and slope are thought to adjust to water discharge (see Métivier et al., 2017 and references therein) such as for instance their width scales with the square root of water discharge (Lacey, 1930). A comparison between the equilibrium slope of the inlet channel and the slopes of disconnected channel observed in the experiment show them to be consistent with the theory (Fig. 8b), although in the case of levee breach a deviation leaned towards lower slopes than the theory whereas it was towards steeper slopes in no forcing scenario. These deviations would confirm that abandonment trigger was dominated by channel entrenchment in the former case and by plug construction in the latter. As our experimental observations are compatible with the Lacey's law and the threshold theory, even in such constrained settings (i.e., fixed wall), they are likely to apply in nature.





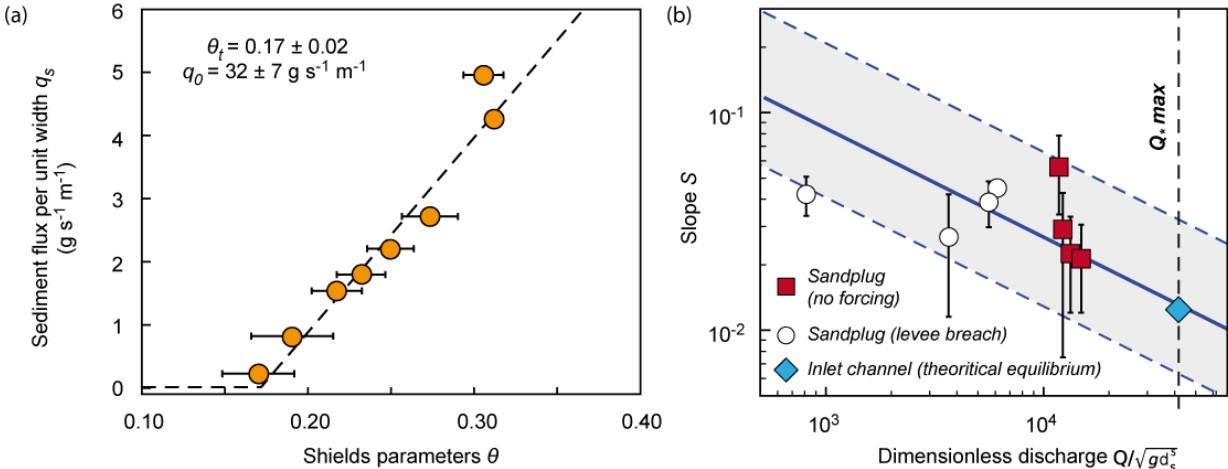

**Figure 8: (a) Transport law. Volumetric flux per unit width, as a function of Shield parameter θ. Dashed line corresponds to Eq. (1) fitted to the data (b) Regime relationship between dimensionless discharge $Q_*$ and slope measured in the experiments. Solid blue line correspond to the threshold theory (Eq. 2, $\theta_t$=0.17 and $C_f$=0.06). Shaded area and dashed lines indicate uncertainty based on varying friction coefficient $C_f$ value.**

An intermediate scale between our flume experiment and natural systems is the irrigation system channels, which have also fixed width. Intakes' plugging is an issue that many water management engineers face, and the question of optimal diversion angle value has been intensively studied. Novak et al. (1990) and Munir et al. (2011) state for instance that a 30 to 45° diversion angle is desirable to limit silting up of the channel whereas a 90° angle is "the least desirable one". This statement is consistent with the present study where channels with high diversion angles are easily plugged and disconnected. Neary et al. (1999) observed sedimentation patterns similar to this study (i.e., a sandbar anchored on the external bank of the deviated channel immediately downstream of the bifurcation) in a 90° lateral intake adjoining the Ohio River, both in the actual intake and numerical simulations. Hence, the first-order observations made in this study seem to apply at larger scales such as larger waterways.

In natural cases, avulsion and disconnection are usually accompanied by the enlargement of the new dominant channel path (Kleinhans et al., 2008) and eventually change of bifurcation angle with time (Bertoldi, 2012). Both impact sandplug construction. In our experiments channel walls were fixed and the dominant flow immediately occupied a channel equal in width to the inlet channel. As the result, disconnection rates were likely faster than when the channel could erode its bank to adjust its shape, similarly to natural systems. Such delay would probably favor the building of larger sandplugs in abandoned channels. To our knowledge, no relationship between diversion angle and sandplug length has been quantified on the field yet. Constantine et al. (2010) found no significant correlation between the incidence angle and the emerged length of the sandplug based on aerial photographs. However, they did find a negative correlation between incidence angle and gravel fill depth below water measured at the apex of abandoned meanders. Such measurements are more comparable to our results (i.e., taking into account the submerged part of the sandplug) and seem to be adequate when investigating on the field for a relationship between incidence angle and bedload deposits. The fact that the subaerial plugs length in abandoned channels is

Earth **Surface**
**Dynamics**
Discussions


not related to bifurcation angles may be due to fine grain deposition over the sandplug after disconnection. More investigations on the field would be required to test this hypothesis.

## 4.5 Sandplug architecture integration to reservoir modeling

The sandplug formation processes observed experimentally in this study are the same independently of the bifurcation geometry and occurrence or absence of levee breach (Figs. 4 and 9). Sandplugs are not simply sediment wedges deposited at

the mouth of the disconnected channels (Fisk, 1947; Allen, 1965) but complex bedload features formed by bars amalgamation (Fig. 9). Sandplugs have a major slope break separating the thicker upstream part that actually plugs the channel and the downstream part, and lateral width variation on the downstream part (Fig. 9). Inherited topography affects the sandplug final architecture (Fig. 6b). For instance, alternate sand bars may be found isolated in the channel. During disconnection the sandplug may rework some of the previously deposited bars to form transitional bars (Figs. 3c and 9).

Together, they form a consistent coarse-grain plug that extends inside the paleo-river path.

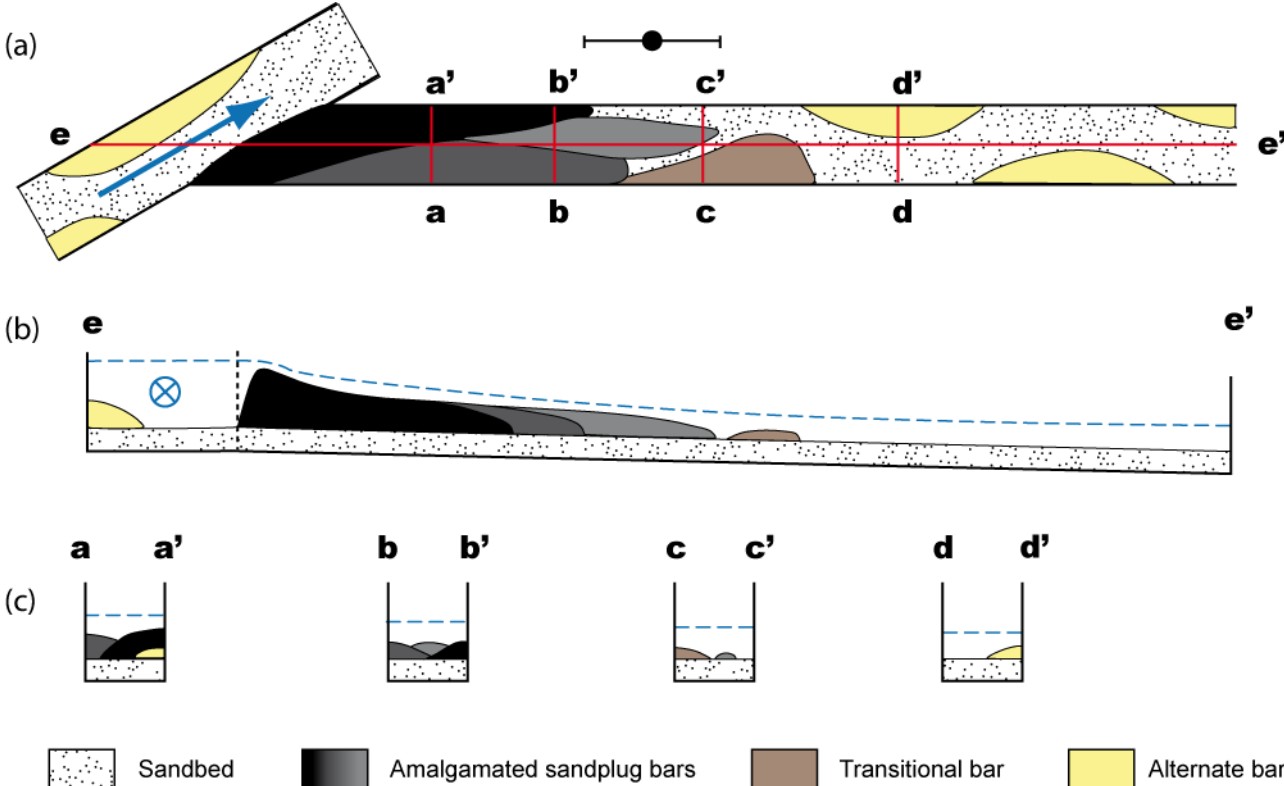

**Figure 9: Conceptual architecture of sandplug derived from experimental observations: overhead view (a), longitudinal profile (b) and transverse sections (c).**

In natural cases, the successive episodes of construction would imply the presence of grain-size variations in the internal

structure of the sandplug. These heterogeneities would include permeability baffles formed at the interface between sandbars during low energy phases and possible erosional surfaces formed during phases of high energy, increasing permeability. In

these cases, the internal structure of a sandplug could be comparable to that of a complex point bar (Deschamps et al., 2012; Cabello et al., 2018) and thus form a good reservoir in itself.

Finally, the presence of bedload deposits in disconnected channels demonstrates how the abandoned channel fills could be connectivity bridges rather than permeability barriers, especially in the upstream part of abandoned channels (Donselaar & Overdeem, 2008). For instance, sandplug lengths vary between 5 and 12 $W$ in the present study (Fig. 7a). The latter is in the range of the wavelength for typical meandering rivers (Williams, 1986) and would allow a connection between about 3-4 point bars. Exploring the ranges of sandplug lengths is therefore a promising research area for reservoir characterization in the future. For instance, the construction and amalgamation of plug bars may be different in curved channel than in the straight channels used in this study.

## 5 Conclusion

Based on the series of experiment designed to force channel abandonment under constant water and sediment discharge, we find that:

(1) Above a diversion angle threshold of 22.5°, discharge partitioning becomes unequal. Disconnection occurs when the diversion angle is greater or equal to 30°.

(2) Sandplug length and volume linearly decrease with the diversion angle.

(3) The incidence angle controls the width of the flow separation zone in the diverted channel. When the incidence angle is high enough, sandplug formation begins in this zone and its presence creates a feedback loop leading to further deposition until disconnection occurs.

(4) The sandplug is a complex structure formed by amalgamated and interconnected sand bars of various lengths, widths, elevation and slope, which may increase the connectivity of fluvial reservoirs, in particular between otherwise isolated point bar deposits.

## 6 Authors contribution

L. Szewczyk and J.-L Grimaud built the flume and designed the experiments with input from I. Cojan. L. Szewczyk carried out all experiments. All authors contributed to writing the manuscript.

## 7 Competing interests

The authors declare that they have no conflict of interest.



## 8 Acknowledgements

We are indebted to Aurélien Baudin, Cyril Leipp, Yasmina Habaoui, David Marquez and Loic Marlot for their help during
the building of the flume. We thank Joël Billiotte, Cyril Castanet, François Métivier, Gerard Salter and Damien Huyghe for
fruitful discussions on this work. The work was supported by the Carnot MINES institute, ref: 2017-1700557. This work is
part of the first author PhD thesis.

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
