# Peer review of "Experimental evidences for bifurcation angles control on abandoned channel fill geometry"

_Earth Surface Dynamics, 2019_

## Referee Comment (RC1) · John Shaw (Referee) · 19 Feb 2020

Review of Szewczyk et al. "Experimental evidences for bifurcation angles control on abandoned channel fill geometry" by John Shaw, University of Arkansas This manuscript describes experiments showing controls on sand plug length and volume at the distributary bifurcation of a river. The experimental design and analysis convincingly show that under these experimental conditions, (a) an increasing diversion angle reduces sand plug length and volume, and (b), that an increased slope in the channel with the depositing sandplug produces larger length and volume. The generalizations to field scale sandplugs is reasonable, although the generalization to the permeability structure of a channel belt is perhaps too far. I find these results to be a thorough and clear scientific advance. Apart from a sprinkling of sentence edits (see below), it is

[Figure]

well written, illustrated, and argued. I think it will make a strong contribution to ESURF. I have sat with this paper for three days, and I can't find a significant flaw. Hence, I recommend it for publishing with the technical below.

Minor comments

Line 8: such as sandplugs

L12: Recommend letting the abstract reader know that this is an inverse relationship.

L13-14: Consider revising these lines so that initiates is not used twice.

L16: lacks *some of* the complexities. Physical experiments are valuable because they include a great deal of complexity.

L17: It is unclear how this paper "improves the realism of fluvial models. Consider revision.

L28: Donselaar and Overeem (here and elsewhere in manuscript).

Equation 2. Is it really $2^{(2/3)}$ in this formulation?

L172: How is it know that velocity was lowest? Was it visual inspection? This is fine, but please say so. Or is the an interpretation due to the presence of the bar? In the later case, I would not necessarily interpret that the velocity was lowest.

L185: Scenarios?

L239: hypothesize

L276: Far above the *bottom* ones. I don't know what bottom adds here.

L292: "Repartition" is used four times in the manuscript, but I do not know the meaning of this word.

L292-3 incidence angle used twice in the same sentence.

L313: explicitly name the relation between width and discharge Lacey's law, if you will

refer to it as such in line 319

L314: disconnected channels

L315-316: a somewhat confusing sentence.

L344: "field investigations" instead of "investigating on the field"

L367: The authors seem to suggest that a channel plug could exted 3-4 point bars, or 3-4 bends into an abandoned channel. This seems like an extremely long way based on my intuition. Are there any field studies that show this type of extension?

---

## Referee Comment (RC2) · Richard Hale (Referee) · 24 Feb 2020

In this study, the authors design a series of flume experiments to test how bifurcation angle and avulsion processes impact the growth of sand plugs. I find this article well written and well defended, with intuitive illustrations and careful explanations. In particular, Figure 7 stands out as particularly informative and including a remarkable amount of information. I agree with the comments made by previous reviewers, and can find no fundamental flaw with their argument or presentation. Indeed, despite careful scrutiny, I could only identify a small handful of suggested grammatical changes. I recommend publication with these minor modifications.

L160 – speed should probably be singular, as it relates to "a sandplug" L326 – "Intake

plugging" L337 – "As a result"

**ESurfD**

Interactive
comment

---

## Author Comment (AC1) · 27 Feb 2020

JS: This manuscript describes experiments showing controls on sand plug length and volume at the distributary bifurcation of a river. The experimental design and analysis convincingly show that under these experimental conditions, (a) an increasing diversion angle reduces sand plug length and volume, and (b), that an increased slope in the channel with the depositing sandplug produces larger length and volume. The generalizations to field scale sandplugs is reasonable, although the generalization to the permeability structure of a channel belt is perhaps too far. I find these results to be a thorough and clear scientific advance. Apart from a sprinkling of sentence edits (see below), it is well written, illustrated, and argued. I think it will make a strong contribution to ESURF. I have sat with this paper for three days, and I can't find a significant flaw.

Hence, I recommend it for publishing with the technical below.

AR: We thank the reviewer for the positive and constructive comments. The effect of sandplug architecture on channel belt permeability is a current focus in the frame of the first author PhD thesis but we acknowledge that the generalization made is a bit too much. We will slightly modify the discussion accordingly.

Minor comments:

JS: Line 8: such as sandplugs

AR: This will be corrected as suggested.

JS: L12: Recommend letting the abstract reader know that this is an inverse relationship.

AR: Thank you for pointing that out, this will be specified.

JS: L13-14: Consider revising these lines so that initiates is not used twice.

AR: This will be done.

JS: L16: lacks *some of* the complexities. Physical experiments are valuable because they include a great deal of complexity.

AR: Indeed. We will do the correction as suggested. What we meant is that in the set up, the system is a bit confined compared to other experiments on delta for example. It is for this reason that the transport law was tested.

JS: L17: It is unclear how this paper "improves the realism of fluvial models. Consider revision.

AR: The relationships shown in Figure 7 could be a step forward to model channel fills in geomorphic and reservoir modeling, with implications on the realism of the topography of the alluvial plain and the facies (∼grain size) distribution of sedimentary deposits. For instance, the models for evolving meandering channels using the Hasegawa-Ikeda-

Parker-Sawai equation (Parker et al., 2011) do specifically model channel fill and some-time assume 100% filling with mud plug. Using a dependence of sandplug length and volume on bifurcation angles will improve these models. In the case of more detailed models, these relations could be tested together with the dynamics of the flow separation zone. We will modify the abstract and discussion to detail these aspects.

JS: L28: Donselaar and Overeem (here and elsewhere in manuscript).

AR: Apologies. This will be corrected throughout the manuscript.

JS: Equation 2: Is it really 2ËĘ(2/3) in this formulation?

AR: Thank you for pointing this out. The correct formulation is indeed 2^(3/2). It will be corrected on the manuscript. This is a typo in the manuscript; the formula used for the calculations was the correct one.

JS: L172: How is it know that velocity was lowest? Was it visual inspection? This is fine, but please say so. Or is this an interpretation due to the presence of the bar? In the later case, I would not necessarily interpret that the velocity was lowest.

AR: We used visual inspection. Some small suspended particles were trapped in the eddies this area prior to bar growth, allowing us an easy determination of the relative flow speed with the other part of the channel. We will point this out in the manuscript

JS: L185: Scenarios?

AR: This will be corrected as suggested.

JS: L239: hypothesize

AR: This will be corrected as suggested.

JS: L276: Far above the *bottom* ones. I don't know what bottom adds here.

AR: This is indeed a typo that will be deleted.

JS: L292: "Repartition" is used four times in the manuscript, but I do not know the

meaning of this word.

AR: It is a mistake and was intended to mean "partitioning". The four occurrences have been corrected to "partitioning".

JS: L292-3: incidence angle used twice in the same sentence.

AR: The second occurrence will be deleted as it doesn't change the meaning of the sentence. "…symmetrical configurations with incidence angles above the 30° threshold (Fig. 3e)."

JS: L313: explicitly name the relation between width and discharge Lacey's law, if you will refer to it as such in line 319

AR: This will be done.

JS: L314: disconnected channels

AR: This will be corrected.

JS: L315-316: a somewhat confusing sentence.

AR: We will rewrite as follow: "Both the equilibrium slope of the inlet channel and the slopes of disconnected channels observed in the experiment are consistent with the threshold theory and remain within the uncertainty range (Fig. 8b). However, in the case of levee breach equilibrium slopes are gentler than the theoretical equilibrium slope, and in the case of no forcing scenarios slopes are steeper."

JS: L344: "field investigations" instead of "investigating on the field"

AR: This will be corrected.

JS: L367: The authors seem to suggest that a channel plug could extend 3-4 point bars, or 3-4 bends into an abandoned channel. This seems like an extremely long way based on my intuition. Are there any field studies that show this type of extension?

AR: This seems indeed to be a very long sandplug extension, especially because we

**ESurfD**
[Figure]

are used to track it using satellite images. When looking under water, the plug deposits are thinning away but do extend further. In the Rhine Delta, some bedload deposits have been shown 3 to 5 meander bends of avulsion channels (admittedly ones with low sinuosity) downstream of the bifurcation point along cross-sections by Stouthamer (2001) and Toonen et al. (2012), although the purpose of their papers was not to study those deposits in particular. We will point these references in the discussion. We will add the following reference: Stouthamer, E. Sedimentary products of avulsions in the Holocene Rhine–Meuse Delta, The Netherlands. Sediment. Geol. 145, 73–92, 2001.

---

## Author Comment (AC2) · 27 Feb 2020

RH: In this study, the authors design a series of flume experiments to test how bifurcation angle and avulsion processes impact the growth of sand plugs. I find this article well written and well defended, with intuitive illustrations and careful explanations. In particular, Figure 7 stands out as particularly informative and including a remarkable amount of information. I agree with the comments made by previous reviewers, and can find no fundamental flaw with their argument or presentation. Indeed, despite careful scrutiny, I could only identify a small handful of suggested grammatical changes. I recommend publication with these minor modifications.

AR: We thank the reviewer for the nice comment.

[Figure]

Minor comments:

RH: L160: speed should probably be singular, as it relates to "a sandplug"

AR: This will be corrected as suggested.

RH: L326: "Intake plugging"

AR: This will be corrected as suggested.

RH: L337: "As a result"

AR: This will be corrected as suggested.

---

## Author Response (AR1)

Manuscript esurf-2019-79: "Experimental evidences for bifurcation angles control on abandoned channel fill geometry" by L. Szewczyk et al.

Response to reviewers

**Reviewer 1: John Shaw**

This manuscript describes experiments showing controls on sand plug length and volume at the distributary bifurcation of a river. The experimental design and analysis convincingly show that under these experimental conditions, (a) an increasing diversion angle reduces sand plug length and volume, and (b), that an increased slope in the channel with the depositing sandplug produces larger length and volume. The generalizations to field scale sandplugs is reasonable, although the generalization to the permeability structure of a channel belt is perhaps too far. I find these results to be a thorough and clear scientific advance. Apart from a sprinkling of sentence edits (see below), it is well written, illustrated, and argued. I think it will make a strong contribution to ESURF. I have sat with this paper for three days, and I can't find a significant flaw. Hence, I recommend it for publishing with the technical below.

We thank the reviewer for the positive and constructive comments.

The effect of sandplug architecture on channel belt permeability is a current focus in the frame of the first author PhD thesis but we acknowledge that the proposed generalization to channel belt permeability structure was likely beyond the experimental results. We have slightly modified the discussion accordingly.

**Minor comments:**

**Line 8: such as sandplugs**

This has been corrected as suggested.

**L12: Recommend letting the abstract reader know that this is an inverse relationship.**

Thank you for pointing that out, this has been specified.

L11-12: "We find that disconnection is possible in the case of asymmetrical bifurcations with high diversion angle ( $\geq$ 30°) and quantify for the first time an inverse relationship between diversion angle and sandplug length and volume."

**L13-14: Consider revising these lines so that initiates is not used twice.**

This has been revised as suggested.

L14: "Sedimentation in this zone induces a feedback loop..."

**L16: lacks \*some of\* the complexities. Physical experiments are valuable because they include a great deal of complexity.**

Indeed. What we meant is that in the set up, the system is a bit confined compared to other experiments on delta for example. It is for the reason why the transport law was tested.

We have inserted the suggested clarification.

L16: Although our setup lacks some of the complexity of natural rivers..."

**L17: It is unclear how this paper "improves the realism of fluvial models. Consider revision.**

The relationships shown in Figure 7 could be a step forward to simulate channel fills in geomorphic and reservoir modeling, with implications on the realism of the topography of the alluvial plain and the facies (~grain size) distribution of sedimentary deposits. For instance, the models for evolving meandering channels using the Hasegawa-Ikeda-Parker-Sawai equation (Parker et al., 2011) do not specifically model channel fill and sometime assume 100% filling with mud plug. Using a dependence of sandplug length and volume on bifurcation angles will improve these models. For instance, Howard (1996) showed that channel pattern is slowed down by the increased bank cohesiveness of mud-dominated oxbows fills and inversely enhanced when banks are dominated by sand lithology. Such behavior strongly influences the topography and thus the resulting deposits' architecture in alluvial plains by widening or narrowing channel belts. Finally, using models including more detailed description of sediment transport, the relations proposed in our study could be tested together with the dynamics of the flow separation zone.

We modified the abstract and discussion to detail these aspects.

L17-18: "Taken into account, these new data would improve fluvial (reservoir) models by incorporating more realistic topography and grain size description in abandoned channels."

L371-377: "Finally, the presence of bedload deposits in disconnected channels demonstrates how the abandoned channel fills could be connectivity bridges rather than permeability barriers, especially in the upstream part of abandoned channels (Larue & Hovadik, 2006; Donselaar & Overeem, 2008). This could also have a significant impact on fluvial river models. Indeed, common models for meandering channels migration (e.g., Parker et al., 2011) usually assume that abandoned channels are 100% filled by mud plug, with consequences on the erodability of alluvial plains and thus channel migration (Howard, 1996). Using a dependence of sandplug length and volume on diversion angles will influence the overall connectivity in reservoir modeling (i) at the scale of channel fills and potentially (ii) at the scale of channel belts."

Due to these changes we added three references to the reference list:

- Howard, A. D.: Modeling channel evolution and floodplain morphology, in Floodplain Processes, edited by Anderson, M. G., Walling, D. E., Bates, P. D., John Wiley & Sons, Hoboke, USA, 15-62, 1996.
- Larue, D.K., Hovadik, J.: Connectivity of channelized reservoirs: a modelling approach. Petrol. Geosci. 12, 291–308, 2006.
- Parker, G., Shimizu, Y., Wilkerson, G.V., Eke, E.C., Abad, J.D., Lauer, J.W., Paola, C., Dietrich, W.E., Voller, V.R.: A new framework for modeling the migration of meandering rivers. Earth. Surf. Proc. Land. 36, 70–86, 2011.

**L28: Donselaar and Overeem (here and elsewhere in manuscript).**

Apologies. This has been corrected throughout the manuscript. (L29, L373)

**Equation 2: Is it really 2(2/3) in this formulation?**

Thank you for pointing this out. The correct formulation is indeed  $2^{(3/2)}$ . It has been corrected on the manuscript. This is a typo in the manuscript; the formula used for the calculations was the correct one.

Equation 2:
$$S = \left[\sqrt{\mu} \left(\frac{\theta_c(\rho_s - \rho)}{\rho}\right)^{\frac{5}{4}} \sqrt{\frac{\kappa[\frac{1}{2}]2^{\frac{3}{2}}}{3C_f}}\right] Q_*^{-\frac{1}{2}}$$

L172: How is it know that velocity was lowest? Was it visual inspection? This is fine, but please say so. Or is this an interpretation due to the presence of the bar? In the later case, I would not necessarily interpret that the velocity was lowest.

We did use visual inspection. Some small suspended particles were trapped in the eddies located in this area prior to bar growth, allowing an easy determination of the relative flow speed within the other part of the channel. We pointed this out in the revised manuscript.

L175: "Visual inspection showed that in this area flow velocity was the lowest..."

**L185: Scenarios?**

This has been corrected as suggested. (L188)

**L239: hypothesize**

This has been corrected as suggested. (L242)

**L276: Far above the \*bottom\* ones. I don't know what bottom adds here.**

This is indeed a typo that has been deleted. (L280)

**L292: "Repartition" is used four times in the manuscript, but I do not know the meaning of this word.**

Apologies for the Frenglish. It was intended to mean "partitioning". The four occurrences (L276, 281, 296 and 313) have been corrected.

**L292-3: incidence angle used twice in the same sentence.**

The second occurrence has been deleted as it doesn't change the meaning of the sentence.

L 296-297: "...symmetrical configurations with incidence angles above the  $30^{\circ}$  threshold (Fig. 3e)."

**L313: explicitly name the relation between width and discharge Lacey's law, if you will refer to it as such in line 319**

This has been specified.

L317: For instance Lacey's law states that river width scales with the square root of water discharge (Lacey, 1930).

**L314: disconnected channels**

The sentence, including this typo, has been rewritten for better clarity (see comment below).

**L315-316: a somewhat confusing sentence.**

It has been rewritten as follow for better clarity:

L318-321: "Both the equilibrium slope of the inlet channel and the slopes of disconnected channels observed in the experiment are consistent with the threshold theory and remain within the uncertainty range (Fig. 8b). However, in the case of levee breach equilibrium slopes are gentler than the theoretical equilibrium slope, and in the case of no forcing scenarios slopes are steeper."

**L344: "field investigations" instead of "investigating on the field"**

This has been corrected.

L353: "More field investigations would be required to test this hypothesis."

**L367: The authors seem to suggest that a channel plug could extend 3-4 point bars, or 3-4 bends into an abandoned channel. This seems like an extremely long way based on my intuition. Are there any field studies that show this type of extension?**

This seems indeed to be a very long sandplug extension, especially because we are used to track it using satellite images. When looking under water, the plug deposits are thinning away but do extend further.

In the Rhine Delta, some bedload deposits have been shown to extend along 3 to 5 meander bends of the avulsion channels (admittedly ones with low sinuosity) downstream of the bifurcation point along cross-sections made by Stouthamer (2001) and Toonen et al. (2012). Although the purpose of their papers was not to study those deposits in particular we used their data to measure these extensions. We will point these references in the discussion.

L379-380: "This suggested length of bedload deposits is consistent with the presence of bedload deposits mapped in low sinuosity avulsion channels by Stouthamer (2001) and Toonen et al. (2012) 3 to 5 meander bends downstream of the bifurcation point."

Due to these changes we added the following reference to the reference list:

Stouthamer, E.: Sedimentary products of avulsions in the Holocene Rhine–Meuse Delta, The Netherlands. Sediment. Geol. 145, 73–92, 2001.

**Reviewer 2: Richard Hale**

In this study, the authors design a series of flume experiments to test how bifurcation angle and avulsion processes impact the growth of sand plugs. I find this article well written and well defended, with intuitive illustrations and careful explanations. In particular, Figure 7 stands out as particularly informative and including a remarkable amount of information. I agree with the comments made by previous reviewers, and can find no fundamental flaw with their argument or presentation. Indeed, despite careful scrutiny, I could only identify a small handful of suggested grammatical changes. I recommend publication with these minor modifications.

We thank the reviewer for the nice comment.

**Minor comments:**

**L160: speed should probably be singular, as it relates to "a sandplug"**

This has been corrected as suggested.

L163: "A sandplug thus formed, albeit at a different speed."

**L326: "Intake plugging"**

This has been corrected as suggested. (L333)

**L337: "As a result"**

This has been corrected as suggested. (L344)

Additionally, a few typos have been corrected throughout the manuscripts.

[revised manuscript text omitted]